# APEX-based proximity labeling in *Plasmodium* identifies a membrane protein with dual functions during mosquito infection

**Jessica Kehrer**[1,2]*, **Emma Pietsch**[1,3,4], **Dominik Ricken**[1], **Léanne Strauss**[1¤a], **Julia M. Heinze**[1¤b], **Tim Gilberger**[3,4], **Friedrich Frischknecht**[1,2]*

**1** Center for Infectious Diseases, Integrative Parasitology, Heidelberg University Medical School, Heidelberg, Germany, **2** German Center for Infection Research, DZIF, partner site Heidelberg, Heidelberg, Germany, **3** CSSB Centre for Structural Systems Biology, Hamburg, Germany, **4** Bernhard Nocht Institute for Tropical Medicine, Hamburg, Germany

¤a Current address: European Molecular Biology Laboratory, Heidelberg, Germany
¤b Current address: Klinikum rechts der Isar, München, Germany
* kehrer.jessica@med.uni-heidelberg.de (JK); freddy.frischknecht@med.uni-heidelberg.de (FF)

**Data Availability Statement:** All relevant data are in the manuscript and its supporting information files.

## Abstract

Transmission of the malaria parasite *Plasmodium* to mosquitoes necessitates gamete egress from red blood cells to allow zygote formation and ookinete motility to enable penetration of the midgut epithelium. Both processes are dependent on the secretion of proteins from distinct sets of specialized vesicles. Inhibiting some of these proteins has shown potential for blocking parasite transmission to the mosquito. To identify new transmission blocking vaccine candidates, we aimed to define the microneme content from ookinetes of the rodent model organism *Plasmodium berghei* using APEX2-mediated rapid proximity-dependent biotinylation. Besides known proteins of ookinete micronemes, this identified over 50 novel candidates and sharpened the list of a previous survey based on subcellular fractionation. Functional analysis of a first candidate uncovered a dual role for this membrane protein in male gametogenesis and ookinete midgut traversal. Mutation of a putative trafficking motif in the C-terminus affected ookinete to oocyst transition but not gamete formation. This suggests the existence of distinct functional and transport requirements for *Plasmodium* proteins in different parasite stages.

## Author summary

Malaria parasites are transmitted by and to mosquitoes. Blocking either of these transmission steps would stop the parasite life cycle. Hence, identification of new proteins that are essential for these steps are a research priority. The genome of the malaria parasite *Plasmodium* contains over 5500 genes, of which over 30% have no assigned function. Transmission of *Plasmodium spp.* to the mosquito contains several essential steps that can be inhibited by antibodies or chemical compounds such as the egress of gametes from host cells, their fusion or the migration of the ookinete. Yet few proteins involved in these processes are characterized, thus limiting our capacity to generate transmission interfering

**Funding:** This work was funded by the Deutsche Forschungsgemeinschaft (DFG, German Research Foundation) – Project-number 240245660 - SFB 1129 (to FF) and SPP 2225 (to FF and TG), Human Frontier Science Program (HFSP) Young Investigator grant RGY066 (to FF) and the European Research Council (StG 281719) (to FF). JK received part of her salary from funding by the DFG (SFB 1129), the HFSP and ERC. EP received part of her salary from funding by the DFG (SPP 2225). The funders had no role in study design, data collection and analysis, decision to publish or preparation of the manuscript.

**Competing interests:** The authors have declared that no competing interests exist

tools. Here, we establish a method to rapidly identify proteins in a specific compartment within the parasite that is essential for establishment of an infection within the mosquito, and identify over 50 novel candidate proteins. Functional analysis of the top candidate identifies a protein with two independent essential functions in subsequent steps along the *Plasmodium* life cycle within the mosquito.

## Introduction

The causative agents of malaria, unicellular *Plasmodium* spp., survive on a complex life cycle between a vertebrate and a mosquito vector. Interruption of the life cycle is a common goal of intervention studies [1–3]. To this end proteins on the parasite surface can in principle be targeted by antibodies or drugs; yet many of these proteins have either not been identified or not been characterized in depth. Life cycle progression of the parasite and surface location of proteins relies on the secretion of specific vesicles, which differ in content and size from one parasite stage to another. Distinct vesicles are important for intracellular stages to escape from their host cells or the oocyst and for extracellular stages to migrate, cross tissue barriers and enter into host cells. When an *Anopheles* mosquito feeds on an infected individual, it takes up infectious sexual parasite stages, so-called gametocytes, during the blood meal. Within the mosquito gut they rapidly differentiate into female macrogametes and male microgametes that both rely on the release of egress vesicles to escape from their surrounding infected red blood cells (iRBCs). Successful and efficient egress is essential for male gametes to fertilize females and form zygotes which develop into ookinetes [4,5]. Ookinetes, like red blood cell-infecting merozoites and liver cell-infecting sporozoites, depend on the discharge of micronemes for motility, which enables ookinetes to pass through the midgut epithelium to form an oocyst [6]. Gametocyte egress vesicles are distinct from micronemes and appear randomly distributed in the non-polarized cell, while micronemes are located at the apical end of the highly polarized migrating parasite forms [7]. Micronemes contain soluble and membrane bound proteins including adhesins, which are needed for attachment of the parasite to host tissue and cells, a prerequisite for motility and invasion. Known micronemal proteins can function in just one (e.g. CTRP, TRAP) or several parasite stages (e.g. GEST, PAT), while others are expressed at multiple stages but appear to only function during one (e.g. MTRAP) [8–12]. Our limited functional understanding of these proteins suggests that additional proteins are likely to play essential roles in parasite egress, adhesion, motility and invasion [5,13,14]. Yet, 'classic' proteomic approaches often detect large numbers of unspecific proteins. For example, density gradient centrifugation to enrich ookinete-derived vesicles containing CTRP, resulted in a list of 330 putative micronemal proteins [13]. This list, however, included several ribosomal and cytoskeletal proteins, which are clearly cytoplasmic proteins. To more specifically investigate vesicular content, we previously used the promiscuous biotin ligase BirA* [15] coupled to the *Plasmodium berghei* gametocyte specific protein MDV/PEG3, which is localized in the lumen of egress vesicles [9]. This allowed enzyme-based proximity labeling of proteins in egress vesicles of living parasites with biotin (BioID). The labeled proteins were purified via streptavidin-coated beads and identified with mass spectrometry. From the list of the obtained proteins we localized several proteins via fluorescent protein tagging to the secretory vesicles of gametocytes and revealed the involvement of MTRAP in gametocyte egress [9]. BirA*-mediated BioID has since been used to identify new components of the parasitophorous vacuole (PV) and parasitophorous vacuolar membrane, the inner membrane complex, nuclear pore complex components, mitochondria, Maurer's clefts interaction partners, schizogony, the

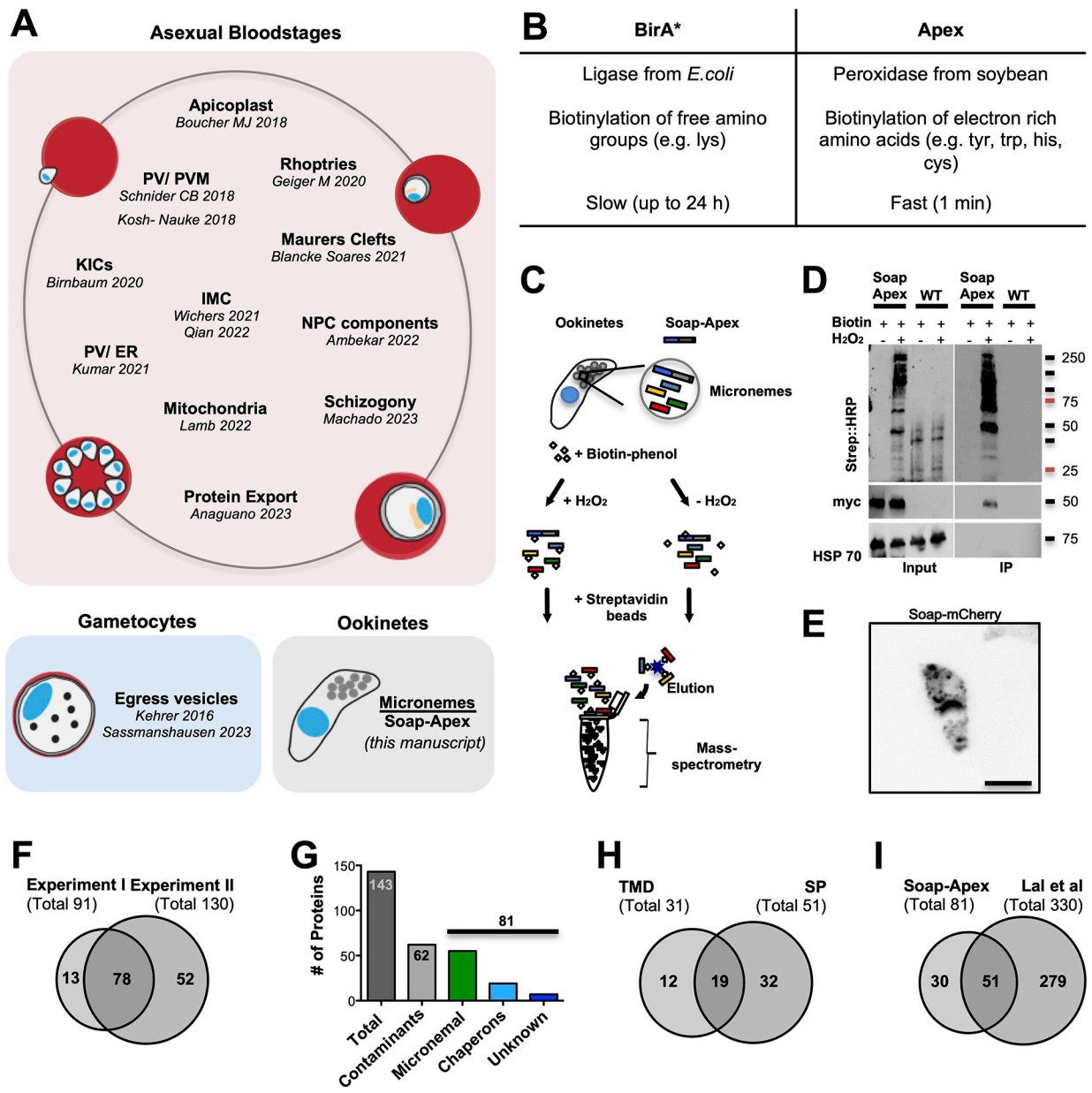

**Fig 1. Proximity labeling in *Plasmodium* and in ookinetes.** (*A*) Overview of BioID approaches performed in *Plasmodium* parasites. (*B*) General comparison of the BirA* and APEX2 based proximity labeling methods. (*C*) Schematic of APEX2 labeling for ookinete micronemes and sample processing for protein enrichment and mass-spectrometry analysis. (*D*) Western blots showing samples of wild type and *SOAP-APEX2* parasites before (input) and after (IP) enrichment of biotinylated proteins, with and without induction of labeling (H₂O₂). (*E*) Live cell imaging of an ookinete expressing SOAP-mCherry. Scale bar: 5 μm. (*F*) Two experiments yielded 91 and 130 proteins with at least 2 peptide hits. 78 proteins were found in both experiments. (*G*) Of 143 identified unique proteins 81 are likely micronemal proteins while 62 were classified as likely contaminants (see S1 Table). (*H*) 31 identified micronemal candidate proteins contained at least one trans-membrane domain (TMD) and 51 a signal peptide (SP), 19 contained both. (*I*) 51 of the proteins identified in our BioID screen were also found by a subcellular fractionation approach [13].

apicoplast, SBP1-mediated protein export, rhoptries and Kelch13 interaction partners of blood stages [16–28]. However, one drawback of the classic BirA*-mediated proximity labeling is the necessary long reaction time of up to 24 h at 37˚C (Fig 1B). Since ookinetes mature within 20 to 24 hours at a temperature of around 21˚C and biotinylation of proteins is restricted to a

short period of time, this did not allow us to identify the vesicular content of ookinete micronemes and motivated us to use the much faster labeling APEX2, a peroxidase isolated from soybean (Fig 1B). In *Plasmodium* blood-stage parasites APEX2 has also been used for the labeling of potential interaction partners of the knob associated histidine rich protein (KHARP) [29] and to visualize de-ubiquitinase UBP1 by electron microscopy [30]. While BirA* catalyzes biotinylation of free amino groups, APEX2-mediated biotinylation occurs within just a minute and prefers electron-rich amino acids such as tyrosine [31,32].

Fusing the micronemal ookinete protein SOAP with APEX2 allowed the robust identification of known micronemal proteins of *Plasmodium* ookinetes as well as a list of unknown proteins. We found 19 proteins that contained both a signal peptide and at least one transmembrane domain. We investigated the 'top hit' in this category, a protein conserved within *Plasmodium* spp. featuring four transmembrane domains and a long divergent extracellular coiled-coil domain by fluorescent tagging, gene deletion and complementation with the *P. falciparum* orthologue. This showed that the protein, termed akratin (from the Greek for powerless/impotent), is expressed in nearly all stages along the parasite life cycle, possibly in the ER and/or secretory pathway and appears essential for microgametogenesis as well as for efficient ookinete migration. Mutation of a putative micronemal targeting motif in the C-terminus of the protein impacted ookinete migration through the midgut epithelium but not gamete egress, suggesting distinct functions or trafficking requirements for targeting proteins to different types of vesicles.

## Results

To identify micronemal proteins of ookinetes we used the microneme-resident protein SOAP (Secreted Ookinete Adhesive Protein, PBANKA_1113400) as bait for proximity labeling. As ookinetes are fully formed within 21 h, but proximity based biotinylation with BirA* may require more than 24 h of incubation, we turned to the faster-acting ascorbate peroxidase APEX2 [32]. SOAP is a small protein of 166 amino acids with a molecular weight of around 21 kDa and important for ookinete-to-oocyst transition [33]. In analogy to our previous approach [9] we generated a parasite line expressing SOAP C-terminally fused to APEX2 and a 3x myc tag (S1 Fig). Fully developed and purified SOAP-APEX2 and wild type control ookinetes were incubated with biotin-phenol for 30 min at room temperature. The cells were then split into two separate populations and biotinylation was initiated in one vial for 1 min using $H_2O_2$ while the second vial served as control (Fig 1C). Samples before and after enrichment of biotinylated proteins with streptavidin-coated beads were tested for successful labeling of proteins by western blot (Fig 1D) and used for proteomic analyses. To monitor the localization of SOAP with fluorescence microscopy an additional parasite line with SOAP fused to mCherry was generated (S2 Fig). Since the protein size of mCherry is comparable to APEX2 this mirrors the strategy employed in the generation of SOAP-APEX2-myc parasites making it more amenable to the visualization of the protein when expressed as a fusion protein. As anticipated, the expression of SOAP-mCherry resulted in a vesicular staining pattern within the ookinete, consistent with previous data (Fig 1E) (Dessens et al., 2003).

In two independent labeling experiments we identified 143 potential micronemal proteins by mass spectrometry of which 78 appeared in both experiments (Fig 1F and S1 Table). Of these 143 proteins, 62 proteins were likely not micronemal such as the cytoskeletal proteins actin and tubulin, ribosomal proteins and those resident in the Golgi. Hence, we classified them as likely contaminants (Fig 1G and S1 Table). Among the remaining 81 putative microneme-resident proteins for which at least two peptides were detected, we found the well-known micronemal ookinete proteins CTRP, WARP, SOAP, chitinase, PSOP17 and PSOP1

[33–37] (S1 Table) as well as aminopeptidase (APP) and protein disulfide isomerase (PDI), which have been localized to vesicles reminiscent of micronemes but not functionally studied in ookinetes [13,36,38]. Furthermore, we identified 19 chaperones and six conserved *Plasmodium* proteins with unknown function, which we listed as likely micronemal candidates (Fig 1G and S1 Table). The top 20 candidate proteins with the most hits (over 20 peptides) contained 19 putative micronemal proteins including four chaperones and only one likely contaminant (alpha tubulin 2) with the three most abundant proteins (99 to 135 hits) being members of the LCCL domain containing protein family (CCp/ Lap) [39]. The 54 candidate proteins with 5 to 20 peptide hits already featured 52% likely false positives and four chaperones while in the 69 proteins with only two to four peptide hits we found a similar number of contaminants, 48%, with the number of chaperones increasing to 11 (S1 Table). Yet, even this fraction contained known or likely micronemal proteins such as PSOP1 [36] and the ookinete surface protein p25 [40,41] (S1 Table).

51 of the down-selected 81 proteins contained an N-terminal signal peptide (SP) and 31 showed at least one transmembrane domain (TMD) with 19 proteins featuring both (Fig 1H). Only 18 proteins did not show any detectable domain as annotated on PlasmoDB (S1 Table). Comparing the 81 proteins with previous data from a cell fractionation study reporting 330 potential micronemal proteins in ookinetes [13] showed 51 common proteins but also 30 novel candidates (Fig 1I and S1 Table). To aid candidate selection we next interrogated the PlasmoGEM database ([42] plasmogem.org) for genes encoding proteins in our list that were targeted by the batch gene-deletion approach [43]. 24 of the 50 genes targeted by PlasmoGEM were reported to be essential for blood stage development, while parasites lacking 16 further proteins were growing significantly slower and 10 were reported as dispensable genes (S3A Fig and S1 Table).

We finally ranked our identified 81 candidates to ookinete RNA Seq data [44] sorted by the number of peptide hits per protein (without taking protein length into account) or total RNA-seq abundance (S3B–S3D Fig). While the known proteins CTRP, SOAP, WARP, chitinase and PSOP1 were the most abundant when sorted according to RNAseq data, it was surprising that these proteins were not appearing among the top hits when sorted according to the number of detected peptide hits.

To select a first candidate protein for further analyses we returned to our 81 likely micronemal candidates (Fig 1G and S1 Table). Among this group we identified 6 conserved *Plasmodium* proteins with hitherto unknown function (Fig 1G and S1 Table) and selected the protein with the highest numbers of peptide hits (16), containing both a SP and at least one TMD. This protein, PBANKA_1105300, contains 455 amino acids and is predicted to feature a signal peptide, four TMDs (as identified through TMHMM and plasmodb.org) and two coiled-coil domains (http://smart.embl-heidelberg.de), suggesting a role in protein-protein interactions (Fig 2A). PBANKA_1105300 was not present in our previous BioID screen from gametocytes and was not probed by PlasmoGEM or present in the study of Lal et al. 2009, but featured in the total sporozoite proteome [14]. Using BlastP (https://blast.ncbi.nlm.nih.gov/Blast.cgi) PBANKA_1105300 was found to be conserved among *Plasmodium* spp. It shares 82% identity with its orthologue in *P. yoelii*, but only 35% and 32% with its orthologues in *P. falciparum* and *P. vivax*, respectively. In contrast to *P. berghei* and *P. falciparum*, the orthologues in *P. yoelii* and *P. vivax* are predicted to contain only two TMDs. Furthermore, due to an N- terminal extension the *P. falciparum* protein contains about two times more amino acids than the *P. berghei* protein. No orthologue was found outside *Plasmodium* spp. (Figs 2B and S4). Transcriptome data of PbANKA_1105300 showed a peak abundance in fully developed ookinetes in comparison with lower expression in gametocytes, asexual stages and sporozoites (Fig 2C) (Otto et al., 2014). Comparing data of male and female gametocytes, expression was found to be higher in males [14,44,45] (Fig 2D).

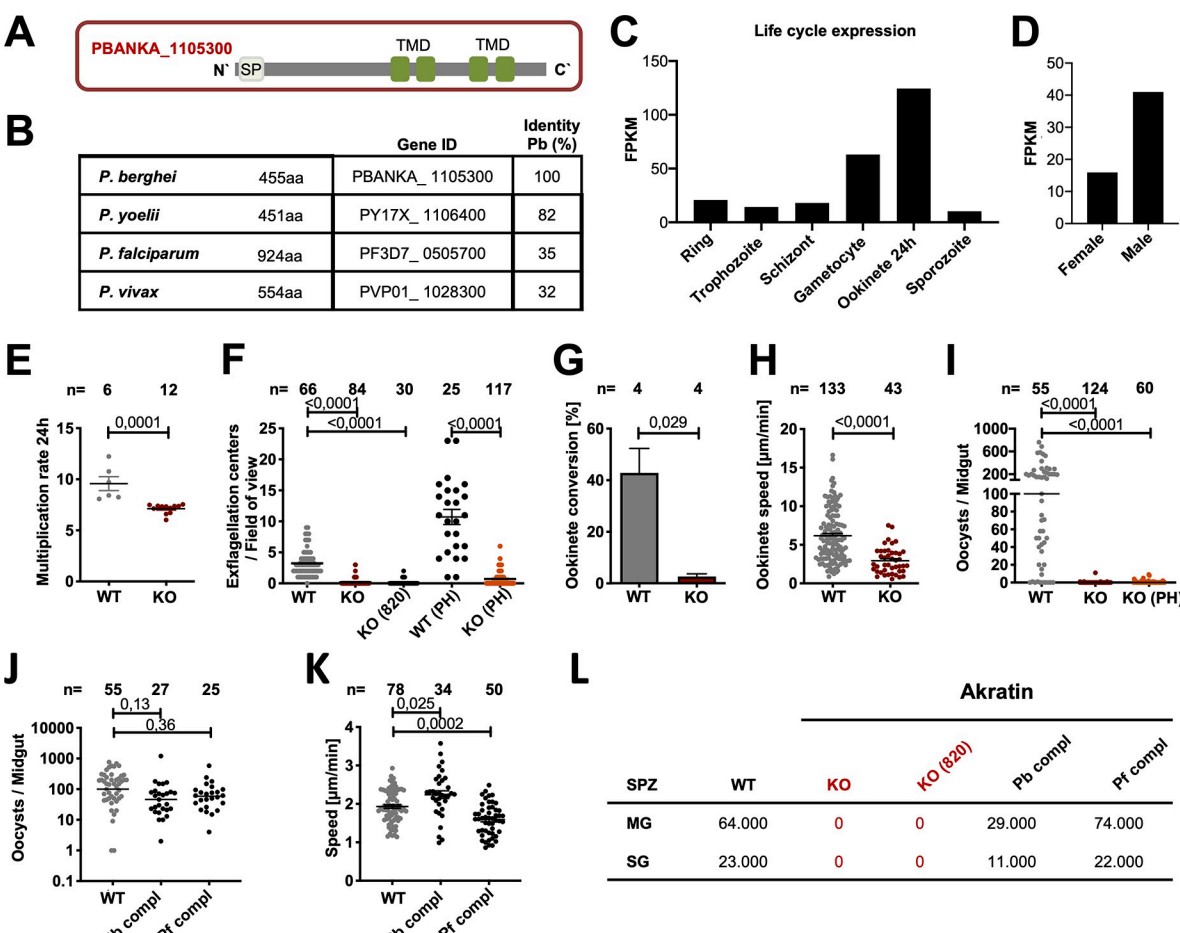

**Fig 2. Akratin is expressed across the life cycle and essential for mosquito infection.** (*A*) Cartoon of the protein features of PbANKA_1105300 in *P. berghei* highlighting the signal peptide (SP) and 4 transmembrane domains (TMD). (*B*) Akratin (PBANKA_1105300) is conserved across *Plasmodium spp*. (*C–D*) *Akratin* expression (RNA-seq values) across the life cycle (*C*) and in male and female gametocytes (*D*); [14,44] (*E*) Blood stage growth rate of *akratin(-)* (KO) parasites in comparison to wild type. Data points represent parasites growing in individual mice. (*F*) Exflagellation of microgametes in the presence and absence of phenylhydrazin (PH) is nearly absent in KO parasites. (*G*) *Akratin(-)* parasites show severely reduced ookinete conversion. (*H*) Speed of gliding *akratin(-)* and wild type (WT) ookinetes. (*I*) Infected *Anopheles stephensi* mosquitoes barely show any midgut infections even in the presence of phenylhydrazin (PH). P-values calculated by Mann Whitney test (E, G, H) and Kruskal Wallis Test with Dunns multiple comparison (F, I). Horizontal bars indicate mean ± SEM. (*J*) Oocyst development of *akratin(-)* parasites complemented with either the *P. berghei* gene or the *P. falciparum* orthologue fused to GFP. Data points represent individual midguts observed between d12-17 post infection. Horizontal line shows median. P-values are calculated using the Kruskal Wallis test followed by the Dunns multiple comparison test. (*K*) Speed of salivary gland sporozoites imaged with a frame rate of 3 seconds. Data points represent average speed of individual sporozoites. Shown is the mean ± SEM. P-values calculated by Kruskal Wallis test followed by Dunns multiple comparison test. (*L*) Sporozoite numbers in midgut (MG) and salivary glands (SG) from infections by wild type, *akratin(-)*, *P. berghei* and *P. falciparum* complementation counted on d17 post mosquito infection. For each counting at least 20 mosquitos were dissected.

### Parasites lacking *PBANKA_1105300* (akratin) fail to transmit to mosquitoes

For functional analysis of PBANKA_1105300, we generated a *P. berghei* parasite line lacking the gene by replacing its open reading frame (ORF) with a selection cassette suited for fluorescent activated cell sorting (FACS) (S5 Fig). During blood-stage development, the resulting polyclonal parasites lacking PBANKA_1105300 exhibited a significantly reduced multiplication rate within 24 hours, decreasing from around 10 in wild-type parasites to seven (Fig 2E). This suggests an important but not essential function during blood stage development. More

strikingly, parasites lacking PBANKA_1105300 showed a strong reduction in male exflagella-tion, which could not be restored in phenylhydrazin (PH)-treated mice (Fig 2F). We hence named the protein akratin (Greek: akrates, for powerless/impotent). *Akratin(-)* parasites con-sequently showed reduced gametocyte-to-ookinete conversion rates (Fig 2G) and a slight reduction of ookinete motility (Fig 2H). Importantly, feeding of mosquitoes on infected mice failed repetitively to yield oocysts, hence revealing a complete block of the transgenic parasites in transmission to the invertebrate host (Fig 2I).

## Complementation with the *P. falciparum* orthologue rescues *akratin(-)* defects

To exclude a phenotype caused by off-target effects we next generated two parasite lines in which we complemented the *akratin(-)* line with a GFP-tagged version of either *P. berghei* or *P. falciparum* (PF3D7_0505700) akratin, before analyzing the observed gene deletion pheno-type in more detail. After recycling the selection cassette of the KO using negative selection (S3 Fig) we inserted the respective gene into the selection marker-free parasites (S6 Fig). Both par-asite lines containing either the original *P. berghei* gene or the *P. falciparum* orthologue C-ter-minally tagged with GFP rescued the defect in gamete egress and the transmission block of the gene deletion mutant. The observed infection rates of mosquitoes as well as oocysts, sporozoite numbers and motility were overall similar to those in wild type infections (Fig 2J–2L). The *P. falciparum* orthologue is thus functional in *P. berghei* even though it has a long N-terminal extension and an overall identity of only 35% (see S4 Fig).

We next investigated the localization of the fluorescent signal in the blood stage and the extracellular forms of the life cycle. This revealed a mostly diffuse signal with some punctae in blood stages of both parasite lines expressing the *P. falciparum* or *P. berghei* akratin-GFP (S7A and S7B Fig). Interestingly we observed a fluorescence signal of *P. berghei* akratin-GFP in rings, trophozoites, schizonts and gametocytes while *P. falciparum* akratin-GFP was only expressed from schizonts onwards. Furthermore, the GFP signal in parasites expressing *P. fal-ciparum* akratin appeared weaker compared to *P. berghei* akratin (S7A and S7B Fig). In non-activated gametocytes the signal was distributed within the gametocyte cytoplasm. No differ-ence between male and female gametocytes could be observed. However, upon activation two distinct populations, most likely males and females could be observed. While in females the protein was distributed in the cytoplasm of the cell, male (determined by a cloudy Hoechst staining) showed one bright spot in addition to a few weak dots at the periphery (S7C Fig). Ookinetes showed a vesicular/endomembranous staining with a prominent peripheral staining for the *P. berghei* akratin-GFP but an exclusively vesicular pattern for the *P. falciparum* akra-tin-GFP (S8 Fig). Sporozoites isolated from the oocysts or salivary glands showed a similar punctate localization for both lines reminiscent of secretory vesicles but distinct from the strong apical signal seen in GFP-TRAP sporozoites [46]. However, when we performed west-ern blot analyses of both parasite lines we only detected a protein band corresponding to the size of GFP. This strongly indicates that the GFP tag is cleaved off. This suggests that the vesic-ular localization of GFP results from a cleavage of the fusion protein within the ER. We observed previously a similar phenomenon with profilin-GFP, which was also cleaved. How-ever, expressing a fusion with a longer linker allowed the expression of uncleaved profilin-GFP [47]. Therefore, we next introduced a longer linker between akratin and GFP. Curiously, mul-tiple transfection attempts to generate *akratin(-)*^PbAkratin-GFP parasites were unsuccessful and we failed to generate the parasite lines. Conversely, the generation of *akratin(-)*^PfAkratin-GFP par-asites, featuring an extended linker, worked fine and showed similar protein localization (S7D Fig). However, western blot analyses again showed that the fusion protein was not expressed

as only soluble GFP could be detected. Hence, we cannot unambiguously determine the localization of akratin. As the GFP was found in an endomembrane/vesicular location, we assume akratin is also localized in the ER and or downstream vesicles.

## Akratin is essential for microgametogenesis

To probe if akratin functions in male or female gametocytes we performed parasite-crossing experiments with the *akratin(-)* parasite line and a parasite line that either produces only fertile males, *p47(-)*, or only fertile females, *p48/45(-)* [48]. As positive control we crossed the *p47(-)* and *p48/45(-)* lines with each other. This control as well as the *akratin(-)* x *p47(-)* cross resulted in ookinete conversion rates of 35% and 42%, respectively, while the *akratin(-)* x *p48/45(-)* cross produced only few ookinetes (3%) (Fig 3A), suggesting that female *akratin(-)* gametocytes are still fully functional, while male gametocytes are strongly impaired.

To investigate if akratin functions in male gametocytogenesis we generated an additional *akratin(-)* parasite line in the 820cl1m1cl1 (RMgm-164) background (S9 Fig). These selection marker-free parasites express RFP in females and GFP in male gametocytes [49]. The ratio between both sexes in the resulting *akratin(-)-820* line did not show any difference between the wild type and the KO (Fig 3B) suggesting that male gametocytes are still normally formed. To investigate at which stage the cells arrest during microgamete formation, we activated the gametocytes and fixed them at different time points. Male gametogenesis involves 3 rounds of rapid DNA replication, which is accompanied by an increase of the size of the nucleus within the first minutes [50]. DNA staining with Hoechst of activated and non-activated cells showed no difference in nuclear size between *akratin(-)* and wild type parasites (Fig 3C), indicating that *akratin(-)* microgametes are activated and replicate their DNA normally. Staining of the red blood cell membrane, however, revealed that it is not degraded in male *akratin(-)* parasites (Fig 3D) providing a possible function for akratin in membrane lysis. Supporting such a function, anti-tubulin antibodies revealed normally formed albeit non motile flagellar axonemes, that appeared trapped within the red cell in *akratin(-)* gametocytes, while wild type gametes readily extended from the cell (Fig 3E).

## *P. falciparum* gametocytes lacking akratin do not exflagellate

To determine whether akratin has a similar function in the human malaria parasite *P. falciparum*, and whether we could tag the protein in this parasite we first tagged its orthologue PF3D7_0505700 endogenously with mScarlet using the SLI system [51] (S10 Fig) and examined asexual blood stage parasites as well as gametocytes by live cell microscopy. While there was no signal in ring stages, in trophozoites a diffuse mScarlet signal could be detected (Fig 4A) which is generally in line with published RNAseq data [52]. The signal is retained in free merozoites after egress (Fig 4A). This suggested an endomembrane localization, possibly the ER and/or other endomembraneous structures. Pfakratin-mScarlet is also expressed in gametocytes. Nevertheless, the localization pattern of Pfakratin-mScarlet appears atypical: it forms bright accumulations that do not resemble any known structure throughout all gametocyte stages. Even after activation for 20 minutes, an unusual localization pattern persists (Fig 4A). Importantly, western blot analysis readily revealed a fusion protein at the predicted size of Pfakratin-mScarlet (Fig 4B).

Next, the Pfakratin locus was targeted for gene deletion using the SLI-based targeted gene deletion (TGD) system [51] in the NF54/iGP2 parasite line [53] (S11 Fig). The NF54/iGP2 line facilitates the production of large quantities of synchronized gametocytes, providing an opportunity to comprehensively investigate the loss of akratin across all gametocyte stages. The examination of the Pfakratin-TGD line revealed no discernible difference in blood stage

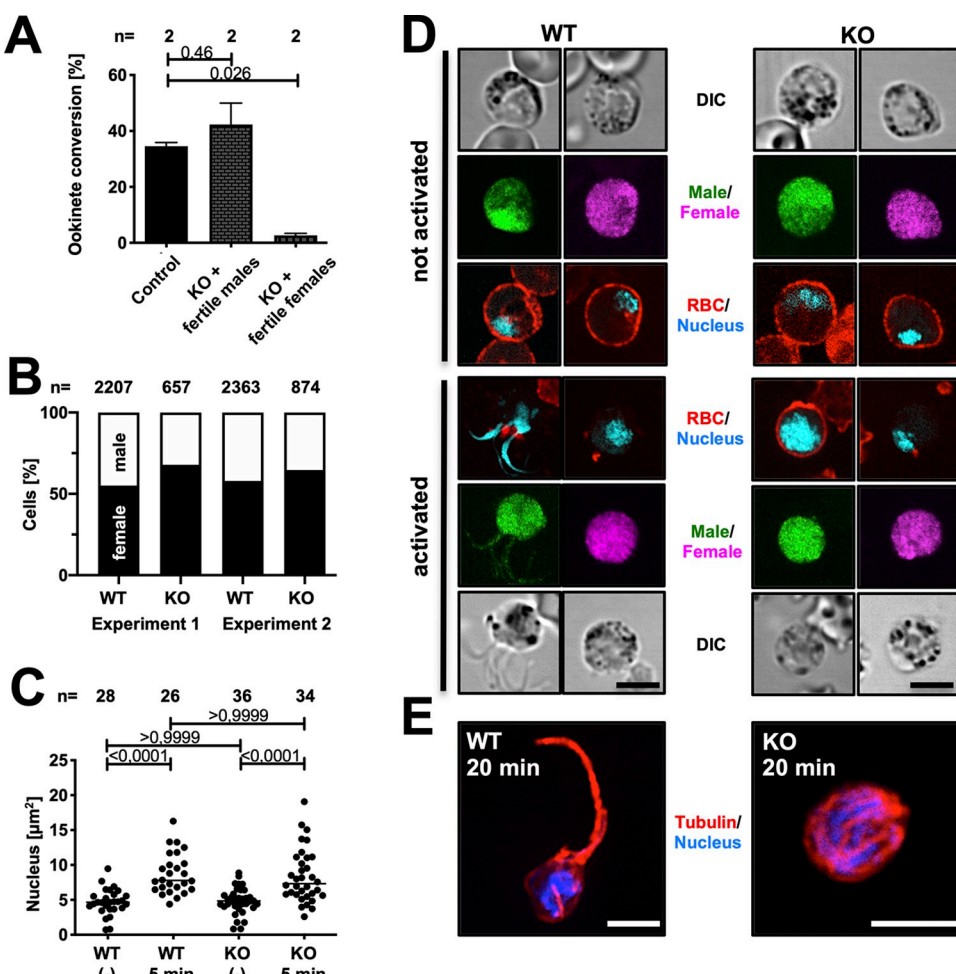

**Fig 3. *Akratin(-)* parasites do not complete microgametogenesis. (*A*)** Crossing of parasite lines deficient in either male or female gametocytes with *akratin(-)* parasite reveals that akratin functions only in males. P-value calculated by One-way Anova with Dunnetts multiple comparison test. Shown is the mean ± SEM. (*B*) The same number of male and female gametocytes are formed in wild type (WT) and *akratin(-)* (KO) parasites lines. (*C*) Nuclear size determined from non-activated (-) or activated (5 min) wild type or *akratin(-)* parasites. Data points represent size of individual nuclei. P-values calculated by Kruskal Wallis Test with Dunns multiple comparison. (*D*) Activation of gametogenesis releases male and female WT gametes but only female *akratin(-)* gametes. Red blood cell (RBC) membrane is stained with Ter119 (red); male parasites are shown in green, females in pink and DNA in cyan. Scale bars: 5 μm. (*E*) 20 minutes post activation *akratin(-)* parasites show microtubule staining reminiscent of assembled axonemes but not egressed gametes, while wild type parasites show the typical axonemal staining of exflagellating gametes. Scale bars: 5 μm.

growth over three cycles. (Fig 4C). After sexual commitment, both, gametocytes of the NF54/iGP2 parental line and the Pfakratin-TGD line show the expected drop in parasitemia due to the removal of asexual stage parasites from the culture (Fig 4D). No morphological differences could be detected during gametocyte maturation (Figs 4E, 4F and S11). Of note, in the NF54/iGP2 background we cannot reliably address a potential role in gametocyte commitment since the commitment transcription factor GDV1 is overexpressed. However, from the results in *P. berghei* we do not expect to see a difference between wildtype and akratin-deficient parasites.

Interestingly, and in line with the results from *P. berghei*, mature stage V gametocytes failed to exflagellate (Figs 4G and S11). Extended incubation of gametocytes in activating conditions could not rescue this deficiency (Figs 4G and S11). To study the egress of stage V gametocytes

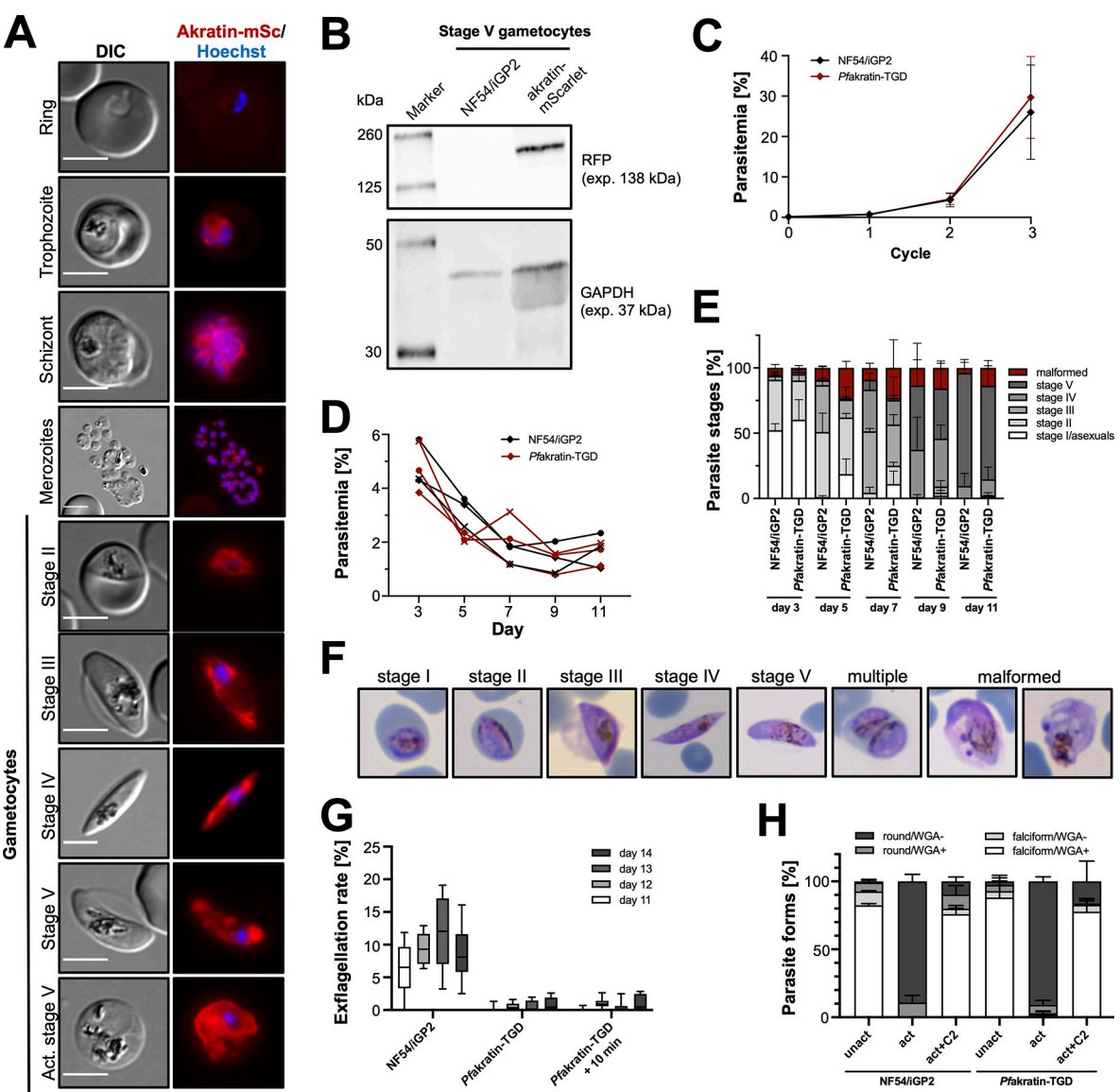

**Fig 4. *P. falciparum* akratin is essential for exflagellation of male gametocytes. (A)** Pfakratin-mScarlet localization in asexual *P. falciparum* blood stages and gametocytes. Nuclei (blue) are stained with Hoechst. For activation, stage V gametocytes were incubated in gametocyte activation medium for 20 min prior to imaging. DIC: differential interference contrast. Scale bars: 5 μm. **(B)** Western blot of Pfakratin-mScarlet stage V gametocytes on day 11 of gametocytogenesis probed with an anti-RFP and anti-GAPDH antibody as loading control. **(C)** Asexual growth of Pfakratin-TGD parasites over three cycles. Parasitemia determined by FACS analysis. After the measurement of cycle 2, the culture was diluted 1:10 to prevent overgrowth. Hence, the parasitemia determined by FACS in cycle 3 was multiplied by 10. Mean ± SD. n = 3. **(D)** Parasitemia during gametocyte development over 11 days. Gametocytemia determined by counting Giemsa-stained thin blood smears. Different symbols indicate the three independent experiments. **(E)** Gametocyte morphology determined by counting Giemsa-stained thin blood smears. Normalized to the total number of single-infected RBC. Mean with SD. n = 3. **(F)** Example images for the gametocyte stages quantified in E. Multiple = multiple-infected iRBC. **(G)** Exflagellation of gametocytes in a 5-min window after 12 min activation. Additional observation time for Pfakratin-TGD parasites as indicated (n = 3). Exflagellation was assessed in technical duplicates in four independent experiments. Interleaved box and whiskers, min to max. **(H)** Egress of stage V gametocytes on day 14 normalized to the total number of gametocytes observed. Mean with SD. n = 3.

in more detail, the RBC membrane was stained using the live-cell dye iFluor 555-WGA. On day 14 of gametocytogenesis, successful activation of both, NF54/iGP2 and Pfakratin-TGD gametocytes, could be observed as indicated by rounding up and loss of the WGA signal upon activation (Figs 4H and S11). Round, WGA-positive iRBC failed to disrupt the RBCM. The

few falciform but WGA-negative parasites were most likely caused by handling. As a control, gametocytes were treated with the egress inhibitor Compound 2 that inhibits the protein kinase G and thus prevents gametocyte activation upstream of the process of rounding up (Figs 4H and S11).

Male and female gametocytes are typically formed in a ratio of 1:3. Due to the lack of exflagellation events observed, we would have expected to observe about one third of activated Pfakratin-TGD gametocytes to remain WGA-positive (WGA+) after activation indicating that the male gametocytes indeed fail to egress while the female gametocyte successfully form mature macrogametes. We found that more than 90% of gametocytes rounded up and lost the WGA signal (Figs 3H and S10) hinting towards a specific exflagellation defect rather than a general defect in activation and egress of gametocytes. Overall, the results strongly suggest that akratin has a male-specific function in microgametogenesis in both, *P. berghei* as well as *P. falciparum* despite minor differences in the observed phenotypes.

## Mutation of a C-terminal motif uncouples akratin function in gametocytes and ookinetes

In mammalian cells the C-terminal sorting sequence YXXϕ (with Y, tyrosine; X, any amino acid and ϕ, hydrophobic amino acid) is important for protein trafficking [54]. A similar motif was also shown to be important for trafficking of the micronemal sporozoite protein TRAP [55], although the sequence is annotated as being located in the trans-membrane domain (Fig 5A). Somewhat similar tyrosine-containing C-terminal motifs (SYHYY and EIEYE) from the *Toxoplasma gondii* adhesin MIC2 were also shown to target the protein to micronemes [56]. We noticed in the C-terminal, possible cytoplasmic domain of *P. berghei* akratin a short sequence (YKKL) representing the described YXXϕ pattern (Fig 5A). However, since the *P. falciparum* orthologue contains a leucine instead of a tyrosine and a conserved glutamic acid to the left we chose to mutate the sequence EYKK instead of YKKL to investigate a possible role of the motif. To this end, we complemented the parasite line lacking akratin with akratin-GFP containing the EYKK motif changed into four alanines (S9 Fig). Akratin-$^{EYKK/AAAA}$-GFP showed similar staining patterns in asexual and sexual parasite stages compared with akratin-GFP (S7 Fig). However, in contrast to the mostly peripheral and vesicular localization of akratin-GFP in ookinetes the akratin-$^{EYKK/AAAA}$-GFP signal was diffusely localized within the cytoplasm (S7 and S8 Figs). Yet interpreting these differences is difficult as western blot analyses indicated again, that the GFP is cleaved and no fusion protein was detected.

Nevertheless, analysis of the mutant line provided an interesting difference to the wild type parasites. Although the exflagellation was somewhat reduced, the mutant readily formed ookinetes at similar rates as wild type which were able to move in a wild type manner (Fig 5B–5D). Strikingly, however, despite their capacity to migrate on glass, the mutant ookinetes failed to form oocysts *in vivo* (Fig 5E). To investigate if this defect is due to the failure of the mutant to penetrate the mosquito midgut or due to a failure in oocyst formation, we isolated midguts 24 hours post infection, washed and stained them for the actin cytoskeleton outlining the epithelial cells and the muscles fibers on the hemolymph-facing side of the midgut (Fig 5F) and searched for ookinetes. This showed half the investigated wild type infected midguts with ookinetes that penetrated the epithelial layer. However, none was found in the mutant (Fig 5F), suggesting that akratin is differentially trafficked in gametocytes and ookinetes and that akratin might be needed on the ookinete surface (or for transport of other proteins to the surface) for interaction with epithelial cells or their traversal. Parasites expressing a mutant akratin appear to lack enough power to enter the epithelium.

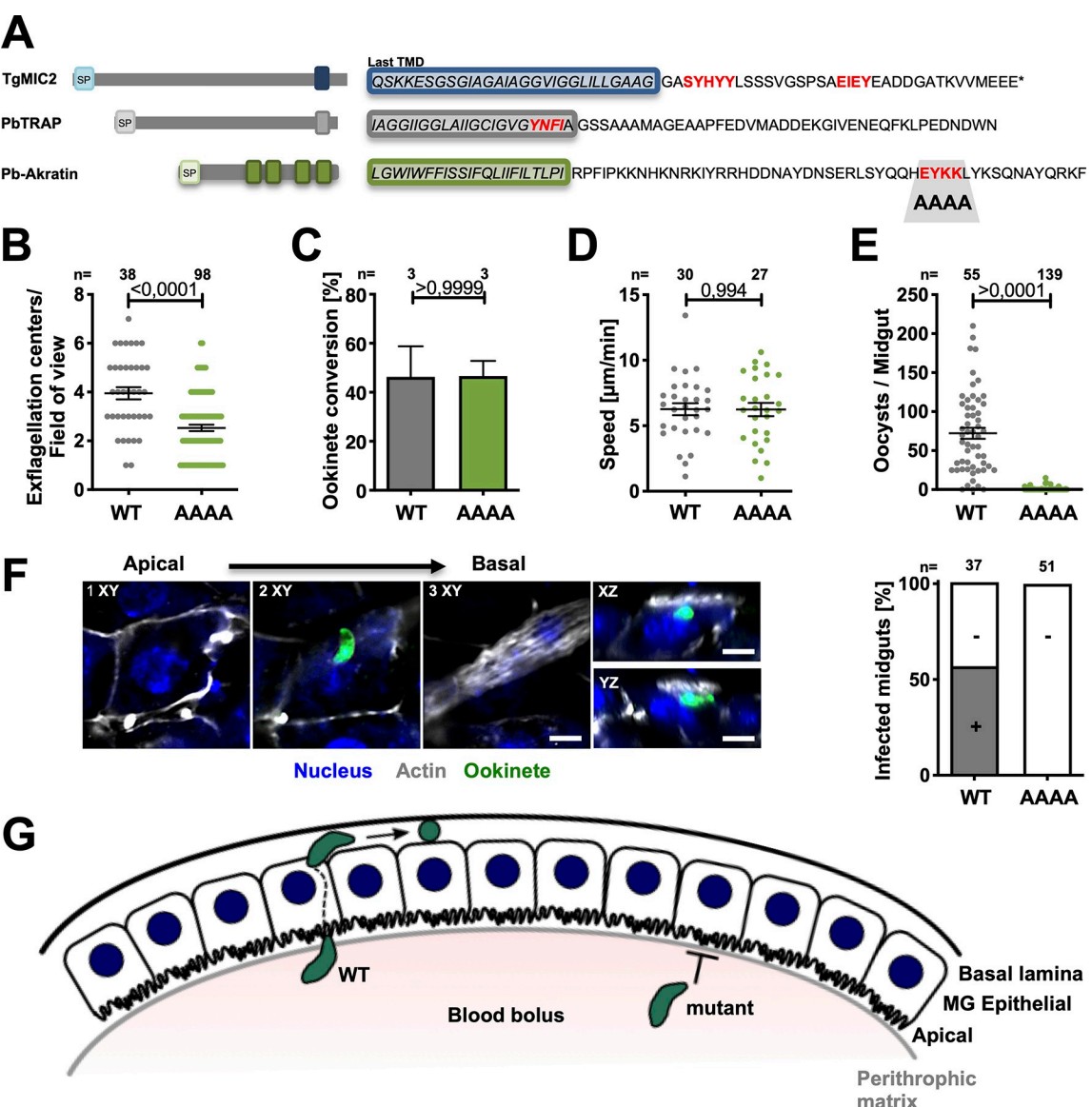

**Fig 5. Mutation of a putative trafficking motif ablates oocyst formation.** (*A*) Cartoon showing signal peptides (SP) and transmembrane domains (TMD) of the *T. gondii* adhesin MIC2, *P. berghei* TRAP and akratin. C-terminal sequences with putative micronemal targeting motifs are highlighted in red. (*B*) Akratin$^{EYKK/AAAA}$ mutant gametocytes can exflagellate, albeit at slightly lower lever than wild type. P-value calculated by Mann Whitney test. Shown is the mean ± SEM. (*C–D*) Ookinete conversion rate (*C*) and motility (*D*) are not affected in the EYKK/AAAA mutant. P-value calculated by Mann Whitney test. Shown is the mean ± SEM. (*E*) Oocyst numbers a dramatically reduced in akratin$^{EYKK/AAAA}$ parasites. P-value calculated by Mann Whitney test. Shown is the mean ± SEM. (*F*) Akratin$^{EYKK/AAAA}$ ookinetes fail to penetrate the mosquito midgut epithelium while more than half of WT midguts showed traversing ookinetes. Confocal images in the indicated views show wild type ookinetes at different positions within the epithelial layer. Scale bars: 5 μm. Graph shows the number of infected and not infected midgut epithelia; no akratin$^{EYKK/AAAA}$ mutants were found in the epithelial cell layer or beyond. (*G*) Cartoon: the path of wild type (WT) and mutant ookinetes from the blood meal across the epithelial layer. The akratin$^{EYKK/AAAA}$ mutant arrests below the peritrophic matrix and WT below the basal lamina.

## Discussion

Using APEX2-based rapid proximity labeling, we aimed to define the content of secretory vesicles in *Plasmodium berghei* ookinetes. With just two experiments we successfully identified all well-known micronemal ookinete proteins as well as over 50 hitherto unknown candidates, some of which can also be found in the sporozoite proteome and might thus also be important

for *Plasmodium* transmission from the mosquito to the mammal. Classifying these candidates into proteins with signal peptides and transmembrane regions allowed us to select a candidate, named akratin, for functional analysis. Deletion of *akratin* resulted in a complete block of parasite transmission into mosquitoes but surprisingly showed a phenotype already in the exflagellation of male gametes nearly abolishing gamete release and hence ookinete formation. Other single or multi-pass membrane proteins such as PAT [46] or MTRAP [9,10] have also been shown to play a role in gamete exit from red blood cells, although their detailed mode of action is not clear. Strikingly, tagging of akratin with GFP in *P. berghei* using two different linkers did not yield a fusion protein. The presence of the GFP in vesicular structures suggests that GFP is cleaved after the protein is reaching the endoplasmic reticulum. This precise localization led us originally to believe that we obtained a fusion protein. As we could not clearly determine the localization of akratin in *P. berghei* parasites, we turned to tagging of akratin with mScarlet in the *P. falciparum* NF54/iGP2 strain [53]. This was straight forward, yielded a fusion protein and suggested an endomembrane localization, possibly the ER and/or other endomembraneous structures in gametocytes (Fig 4A). However, we could not reliably get ookinetes for further localization studies in this parasite stage. The next step in understanding the function of akratin would be to use anti mScarlet antibodies and perform immunoprecipitation analysis to find potential interaction partners. Additional experiments are also needed to determine the localization of akratin in ookinetes. It would be valuable to determine if akratin is localized solely in the endoplasmic reticulum or if it is indeed present in micronemes. As more and more proteins are discovered to have dual functions in gametocytes and microneme-containing motile parasite stages (e.g. GEST and PAT) it will also be interesting to get further insights into this functional relation. Fusion of proteins with fluorescent markers has a drawback as the fluorescent protein usually needs time to fold. Hence, in some of the reported protein localization studies only a later localization is reported. For example, GFP-TRAP is localized nicely in micronemes of *P. berghei* sporozoites, but anti-GFP antibodies also reveal the protein in the ER [46]. Clearly, the GFP-TRAP fusion protein is trafficked through the ER but the GFP is only fully matured to fluoresce once the protein is packaged into the secretory vesicles. This might also hamper a future analysis of akratin. It is hence, currently not clear, if akratin is found in the vicinity of SOAP in micronemes or in the early secretory pathway.

Interestingly, akratin also showed an important function in ookinetes, revealed by the mutation of a putative C-terminal trafficking motif in *P. berghei*. This mutation had no detrimental effect on akratin function during gamete formation. Ookinetes still formed normally and were also equally motile as wild type control parasites on glass, yet they failed to penetrate the midgut epithelium and thus could not develop into oocysts. How can ookinetes still move *in vitro* but not migrate *in vivo*? One possibility is that they fail to bind a key receptor and hence do not signal for a potential switch from migration to invasion. Or the mutant ookinetes do not produce enough force and thus cannot penetrate through the peritrophic matrix that builds around the digesting blood meal or enter the epithelium. We recently observed a similar effect when replacing the essential ookinete adhesin CTRP with the sporozoite adhesin TRAP [11]. Without CTRP ookinetes do not glide on glass and also do not enter the epithelium [34,57,58]. Strikingly, the TRAP expressing ookinetes moved on glass just fine but did not form oocysts, suggesting that *in vitro* motility is not a perfect predictor for *in vivo* migration [11]. Indeed, several mutants generated in our laboratory show a similar disconnect for sporozoites and their capability to enter into salivary glands. Some show a strong defect in gliding on glass, yet enter normally into salivary glands [47], while others migrate almost normally but do not enter efficiently into glands [59,60]. These mutants show mutations in actin or actin-binding proteins and hence should not affect receptor-ligand interactions, although we speculate that receptor-ligand interactions might cause a rearrangement of actin, which could feed

back to modulate the receptor-ligand binding strength [61]. A similar complex outside-in to inside-out signaling might well play a part in the complex journey of the ookinete, which needs to first escape the blood meal, cross the peritrophic matrix, migrate along and through the epithelial cell layer and arrest below the basal lamina, a journey certainly benefiting from fine-tuned sensing of the environment [62,63]. Another possibility for the observed defect of the akratin mutant in penetration of the epithelium would be that the protein protects the ookinete from the onslaught of the vertebrate or mosquito defense proteins such as TEP1 [64]. Currently we can't be sure of the exact function of akratin, also due to the lack of clarity of its localization. Instead of having a direct function, akratin might also be indirectly involved in processes in the secretory pathway that leads to a disbalance of secreted factors important for trans-tissue migration.

In conclusion, we established a new assay for rapid biotinylation in malaria research, generated a list of putative micronemal proteins in ookinetes, which might harbor new transmission blocking candidates and identified a protein, akratin, with dual roles in gamete formation and ookinete migration. While the discovery of akratin opens interesting biological questions the use of rapid BioID will provide an important tool for discovery along the complex *Plasmodium* life cycle.

## Materials and methods

### Generation of transfection plasmids

*SOAP-APEX2*: The *APEX2-myc* sequence flanked with the restriction sites BamHI and XbaI was ordered from geneart (Regensburg, Germany) and ligated into a vector containing SOAP-BirA* (kind gift from Gunnar Mair) exchanging the bira* sequence. The resulting plasmid pL8 was linearized with HpaI prior transfection for single crossover integration into wild type parasites (S1 Fig).

*akratin(-)*: The 3'UTR of PbANKA_1105300 was amplified using primers JK67 and JK68 and inserted into a plasmid containing the recyclable yFCU/ hDHFR selection cassette and GFP expressed under the HSP70 promoter digested with NotI and SacII. The 5'UTR of PbANKA_1105300 was amplified using primers JK65 and JK66 and inserted into the plasmid using KPNI and HindIII. The resulting plasmid pL28 was linearized with NotI and SacII prior transfection for double crossover integration (S3 Fig).

Marker free *akratin(-)*: The drinking water of mice was supplemented with 2 mg/ml 5-FC. Clonal parasites which looped out the selection cassette, were obtained by limiting dilution (S3 Fig).

*P. berghei akratin-gfp* complementation: The 5'UTR together with the entire ORF of PbANKA_1105300 was amplified from wt gDNA using primers JK66 and JK152 and inserted into the pL28 plasmid using KpnI and NdeI leading to the replacement of the selection marker. A TgDHFR selection cassette was amplified using primers JK153 and JK154 and inserted between the GFP and 3'UTR using NotI and EcorV resulting in plasmid pL59. For transfection of parasites via double crossover the plasmid was digested with KpnI and SacII (S4 Fig).

*P. falciparum akratin-gfp* complementation: The ORF of PF3D7_0505700 was amplified using primers JK171 and JK172 and inserted into pL28 downstream of the *P. berghei* 5'UTR using HindIII and NdeI leading to the replacement of the selection marker. The TgDHFR selection cassette was amplified using primers JK153 and JK154 and inserted between the GFP and 3'UTR using NotI and EcorV resulting in plasmid pL75. For transfection of parasites via double crossover the plasmid was digested with KpnI and SacII (S4 Fig).

*P. berghei akratin$^{EYKY}$/$^{AAAA}$*: The 5'UTR together with the entire ORF was amplified from wt gDNA using primers JK66 and a reverse Primer introducing the mutations and inserted

into the pL59 plasmid using KpnI and NdeI resulting in pL84. For transfection of parasites via double crossover the plasmid was digested with KpnI and SacII and transfected into negative selected KO parasites (S6 Fig).

### Generation of transgenic parasite lines

Transfection of linearized plasmids was performed as previously described [65]. After electroporation of schizonts, positive selection was performed using pyrimethamine. All resulting lines were cloned through limiting dilution except *SOAP-APEX2* and *akratin(-)*. *SOAP-APEX2* parasites still contain a small amount of wild type parasites that did not interfere with BioID, while the *akratin(-)* was obtained through FACS sorting as follows: 2 drops of blood from the mouse tail vein with a parasitemia of maximum 0.3% were diluted in 1.5 ml of RPMI medium. To avoid clustering of cells the sample was drained before analysis, with a 40 μm cell strainer. For detection, cells were analysed with a 488 nm laser in combination with a 527/32 nm filter for the GFP signal. To set up gates for sorting, the voltage of forward scatter (FSC) and sideward scatter (SSC) were adjusted to mainly gate on erythrocytes and a singlet discrimination was performed. Cells were sorted in the purity mode with the lowest flow rate at room temperature using a BD FACS Melody. As sheath fluid PBS was used. Sorted cells (hundred events) were immediately injected i.v. into naïve mice. No wild type could be detected in the resulting polyclonal *akratin(-)* population.

### *In vitro* ookinete cultures and proximity labeling

20 million blood stage parasites were injected i.p. into a naïve NMRI mouse treated with 100 μl phenylhydrazin (6 μg/ml) 24 hours pre-transfer. 3 days post infection the drinking water was supplemented with sulfadiazin (20 mg/l) to reduce asexual blood stages. 4 days post infection the blood was harvested by cardiac puncture to set up ookinete cultures. The blood of one mouse was added to 10 ml of ookinete medium (RPMI containing 25 mM Hepes and 300 mg/l glutamine, 10 mg/l hypoxanthine, 50 000 units/l penicillin, 50 mg/l streptomycin, 2 g/l NaHCO$_3$, 20.48 mg/l xanthurenic acid, 20% FCS; pH 7.8) at 19˚C and incubated for 21 hours. Fully developed ookinetes were purified on a 63% Nycodenz cushion and the remaining RBCs were lysed with 170 mM NH$_4$CL for 5min on ice.

Cells were washed once with PBS and resuspended in 1 ml of ookinete medium containing biotin phenol (182 μg/ml). After incubation for 30 min at RT cells were split into 2 tubes. One was used as control and in one biotinylation was activated by adding 5 μl of 100 mM H$_2$O$_2$. After 90 seconds the reaction was inactivated by adding 500 μl of 2x quencher solution (20 mM sodium ascorbate, 20 mM Trolox, 10 mM NaN$_3$ in PBS). After washing the cells two times with 500 μl 1x quencher solution samples were then lysed with RIPA buffer (50 mM Tris pH8, 1% Nonidet P-40, 0.5% Na-deoxycholate, 0.1% SDS, 150 mM NaCl, 2 mM EDTA). 20% of the lysate was kept as input control and biotinylated proteins were enriched using streptavidin coated beads at 4˚C overnight as described [15]. Elution of proteins was done with a buffer containing 30 mM biotin, 2% SDS, 160 mM NaCl and 6 M Urea followed by mass-spectrometric analysis as described below.

### Protein analysis and proteomics

Western blots were performed using the following antibodies: anti-myc antibody (Roche 0.4 mg/ml, 1/1000) followed by incubation with goat anti mouse HRP (BioRad 1/10000), Streptavidin- HRP (Sigma, 1/500) or mouse anti-HSP70 followed by incubation with goat anti-mouse HRP. Mass spectrometric analysis was performed at the CellNetworks Core Facility for Mass

Spectrometry and Proteomics of the ZMBH (Zentrum für Molekularbiologie der Universität Heidelberg) as previously described [9].

## Immunofluorescence and light microscopy

After fixation of the parasites with 4% PFA at 4˚C for at least overnight, cells were permeabilized with 0.5% TritonX-100 for 10 min. Staining of the red blood cell membrane was performed with the anti-Ter119-Alexa647 antibody (Biolegend, 0.5 mg/ml; 1/1000) for 1 hour. Cells were washed twice with PBS, and resuspended in PBS containing Hoechst for observation under the microscope.

For staining of microtubules cells were fixed in 2% paraformaldehyde/0.05% glutaraldehyde in microtubule stabilizing buffer (15 g/l PIPES, 1.9 g/l EGTA, 1.32 g/l $MgSO_4 \cdot 7H_2O$ and 5 g/l KOH in water; pH 7.0 with KOH). Staining was performed using an a-tubulin antibody (Sigma clone DM1A, 1/ 500) for 1 hour, followed by an anti-mouse 647 (1/ 500) for 1 hour. Images were either taken on a Zeiss 200 M Axio-Observer widefield (63x; NA 1,4) or Nikon/ PerkinElmer spinning disc confocal (100x, NA 1,4) microscope. Image processing was performed with ImageJ/FIJI [66].

## Ookinete motility and midgut traversal

*In vitro* ookinete cultures were performed as described above. To observe motile ookinetes one drop of cell suspension was transferred onto a microscope slide and covered with a thin cover glass to simulate a confined environment. Movies were taken with a frame rate of 20 seconds for 10 minutes. Speed was analyzed using the manual tracking plugin in FIJI.

For *in vivo* analysis, 20 million blood stage parasites were injected i.p. into a naive Swiss CD1 mouse treated with phenylhydrazin 24 hours prior to transfer. 4 days post transfer about 20 female *Anopheles* mosquitoes were allowed to feed for about 15 minutes. Blood filled midguts were isolated after 24 hours as described previously [67]. Midguts were fixed in cold 4% paraformaldehyde (PFA) for 45 seconds and washed with cold PBS. The epithelial cell layer was then opened longitudinally using two needles to carefully remove the blood bolus followed by a second fixation with 4% PFA overnight.

To better visualize traversed ookinetes an additional immunofluorescence assay was performed. To do so cells were permeabilized with 0,5% Triton X-100 for 30 minutes, incubated with a rabbit anti GFP antibody (abfinity, 0,4μg/μl, 1/40) for 2 hours and after washing incubated with an anti-rabbit 488 secondary antibody plus the addition of Phalloidin TRITC (1mg/ml, 1/500) and Hoechst (10mg/ml; 1/1000) for another 2 hours. Midguts were transferred onto a microscope slide and images were taken on a Nikon spinning disc microscope using either a 40x (N.A. 0,65) or 60x (N.A. 1,4) objective.

## *P. falciparum* cell culture

*P. falciparum* parasites were cultured in 0+ or B+ human RBCs (transfusion blood, Universitätsklinikum Eppendorf, Hamburg) at 5% hematocrit using RPMI 1640 medium supplemented with 25 mM HEPES, 100 mM hypoxanthine, 2 mM choline, and 24 mM sodium bicarbonate and complemented with 0.5% Albumax II (Life Technologies). For the experiments in this manuscript, the NF54/iGP2 parasite line [53] was used. As described previously [53], NF54/iGP2 lines were cultured in the presence of 2.5 mM D-(+)-glucosamine hydrochloride (GlcN) to maintain the glmS ribozyme active during routine parasite propagation. Cultures were maintained at 37˚C in an atmosphere of 94% nitrogen, 5% carbon dioxide and 1% oxygen.

For transfection of constructs, Percoll (GE Healthcare)-enriched synchronized mature schizonts were electroporated with 50 μg of plasmid DNA using a Lonza Nucleofector II device [68]. Transfectants were selected for plasmid uptake and episomal expression in medium supplemented with 3 nM WR99210 (Jacobus Pharmaceuticals). For generation of stable integrant cell lines, WR-selected parasites were grown in the presence of 400 μg/ml Neomycin/G418 (Sigma) to select for integrants carrying the desired genomic modification as described previously [51]. Successful integration was confirmed by diagnostic PCR using the FIREpol DNA polymerase (Solis BioDyne).

## Cloning of pSLI constructs

For the generation of the SLI-based construct pSLI-Pfakratin-TGD, 608 bp immediately downstream of the start ATG of the akratin gene (PF3D7_0505700) were amplified by PCR using the primers PF3D7_0505700-TGD-fw/PF3D7_0505700-TGD-rev and cloned into the pSLI-TGD vector [51] using NotI/MluI. The amplified region served as homology region for the single-crossover integration.

For the generation of the SLI-based tagging construct pSLI-Pfakratin-mScarlet, the C-terminal 887 bp region of the akratin gene was amplified by PCR using the primers PF3D7_0505700-mSc-fw/PF3D7_0505700-mS-rev and cloned into the pSLI-GFP vector [51] using NotI/MluI. The GFP cassette was exchanged for an mScarlet-encoding cassette that was amplified from SP-mScarlet [69] using the primers mSc-fw/mSc-rev and ligated into the final vector using AvrII/SalI restriction sites.

All PCRs for plasmid generation were performed using the Phusion High-Fidelity DNA Polymerase (New England Biolabs). All plasmid sequences were confirmed by Sanger sequencing.

## Gametocyte maturation and egress assays

Sexual commitment and gametocytogenesis were induced as described before [53]. In brief, GlcN was removed by washing a synchronous ring stage culture (2–4% parasitemia) in medium to stabilize the gdv1-gfp mRNA thereby driving sexual commitment (= day -1). After reinvasion (day 1), culture medium containing 50 mM GlcNAc was added to the ring stage progeny and changed daily for six days to eliminate asexual parasites [70]. From day 7 onwards, gametocytes were cultured in culture medium containing 0.25% Albumax II / 5% human serum instead of 0.5% albumax. Gametocyte stages and gametocytemia were monitored by visual inspection of Giemsa-stained thin blood smears.

For stage quantification, pictures of Giemsa slides were recorded using a 63X/1.30 oil objective on a Leica DM 2000 LED microscope equipped with a Leica DMC 2900 camera. Erythrocyte numbers were then determined using the automated parasitemia software (http://www.gburri.org/parasitemia/, >1800 RBCs/sample) and the number of the different gametocyte stages was counted manually. For normalization, the stage counts were normalized to the number of single-infected RBC.

On day 11 until day 14, exflagellation of gametocytes was quantified. To do so, 500 μl of culture were spun down and resuspended in gametocyte activation medium (culture medium but with 20% human serum instead of 0.5% Albumax II, supplemented with 100 μM xanthurenic acid, pH 8.0). After 12 min incubation at 26˚C, exflagellation events and number of gametocytes were counted using a 40X/1.30 oil objective at the Leica DM 2000 LED microscope over the course of 5 min. The exflagellation rate was calculated as the number of exflagellation events per total number of gametocytes examined.

To study egress of both male and female gametocytes, a live cell staining was performed: The RBCM was stained using iFluor 555-Wheat Germ Agglutinin (WGA) Conjugate (5 μg/ml) and Hoechst (5 μg/ml) in culture medium for 15 min at 37˚C. Subsequently, samples were washed in prewarmed medium and kept at 37˚C until activation/imaging. For the non-activated control, resuspended culture was taken for live cell microscopy. For the activated sample, iRBCs were resuspended in gametocyte activation medium and incubated for 20 min at 26˚C before imaging. As a control, activation was also performed in presence of 1 μM Compound 2.

## Growth analysis of pSLI-based TGD parasite line

Schizont stage parasites of the parental line NF54/iGP2 [53] as well as the *Pf*akratin-TGD line were isolated by enrichment on 60% Percoll and incubated with uninfected RBCs for 3 hours to allow rupture and invasion. Parasites were then treated with 5% sorbitol to remove residual schizonts, resulting in a synchronous ring stage culture within a 3-hour window. For growth analysis, these synchronous ring stage cultures were allowed to mature into trophozoites (= day 1, cycle 0). Parasitemia was then determined by flow cytometry and adjusted to 0.1%. Parasitemia was monitored over three subsequent cycles and analyzed by flow cytometry at day 3, 5, and 7. After analysis on day 5, cultures were diluted 10-fold into fresh RBCs to prevent overgrowth. Medium was changed daily.

Flow cytometry-based analysis of growth was performed as described previously [71]. In brief, 20 μl resuspended parasite culture were incubated with dihydroethidium (5 μg/ml, Cayman) and SYBR Green I dye (0.25 x dilution, Invitrogen) in a final volume of 100 μl medium for 20 min at RT protected from light. Samples were analyzed on a ACEA NovoCyte flow cytometer. Single RBCs were gated based on their forward and side scatter parameters. For every sample, 100,000 events were recorded and parasitemia was determined based on SYBR Green I fluorescence.

## *P. falciparum* fluorescence microscopy

The pSLI-*Pf*akratin-mScarlet parasite line was imaged on a Leica D6B fluorescence microscope, equipped with a Leica DFC9000 GT camera and a Leica Plan Apochromat 100x/1.4 oil objective, after adding Hoechst at a final concentration of 5 μg/ml. For each experiment, all images were acquired at identical settings. For clarity, the images were later processed for brightness and contrast using ImageJ.

## Western blotting

For western blot analysis, mature schizonts arrested with the egress inhibitor Compound 2 (1 μM; kindly provided by S. Osborne (LifeArc) and stored as a 10 mM stock in DMSO at −-80˚C) or stage V gametocytes were enriched on 60% Percoll, washed in medium and lysed in western blot lysis buffer (0.5x PBS/4% SDS/0.5% Triton X-100) containing complete protease inhibitor cocktail (Roche). After addition of SDS sample buffer, samples were boiled for 5 minutes at 95˚C before loading them on a 12% gel for SDS-PAGE. Proteins were transferred to nitrocellulose membranes. Membranes were then blocked in 5% milk in TBS (50 mM Tris/150 mM NaCl, pH 7.5) containing 0.05% Tween 20 before staining with rat anti-RFP mAb 5F8 (Chromotek, diluted 1:1,000) primary antibody at 4˚C overnight. After extensive washing in TBS containing 0.05% Tween 20, membranes were incubated with IRDye 800CW-conjugated goat anti-rat antibody (Licor, diluted 1:10,000) in blocking buffer for 1 hour at room temperature. As loading control, membranes were probed using mouse anti-GAPDH [72] (1:2,000) or rabbit anti-BIP [73] (1:2,000) and IRDye 680RD Goat anti-Rabbit or anti-Mouse IgG

Secondary Antibody (Licor, diluted 1:10,000), respectively. Antibody binding was visualized using the Licor Odyssey Fc system.

## Supporting information

**S1 Fig. Generation of SOAP-APEX2 parasites via single homologous recombination.** The cartoon shows the cloning strategy and primers used for genotyping with amplicon sizes of the resulting transgenic line indicated. For primer sequences please see S2 Table. Plasmid contains as resistance marker the dehydrofolatereductase/thymidine-synthase from *Toxoplasma gondii* (*Tgdhfr/ts*) (grey). Note that the second copy of *soap* lacks the ATG and should not be expressed.
(TIFF)

**S2 Fig. Generation of SOAP-mCherry parasites via single homologous recombination.** The cartoon shows the cloning strategy and primers used for genotyping with amplicon sizes of the resulting transgenic line indicated. For primer sequences please see S2 Table. Plasmid contains as resistance marker the dehydrofolatereductase/thymidine-synthase from *Toxoplasma gondii* (*Tgdhfr/ts*) (grey). Note that the second copy of *soap* lacks the ATG and should not be expressed.
(TIFF)

**S3 Fig.** (***A***) 50 of the 81 micronemal candidate proteins were already investigated by the Plas-moGEM screen of which 24 were classified as essential, 16 conferred slow growth upon deletion and 10 were dispensable. (***B-C***) RNA-seq abundance (y-axis) of the identified micronemal candidate proteins [44] with known proteins (green) and the top candidate, PBANKA_1105300 (red) marked sorted (x-axis) according to RNA-seq data (**B**) and by peptide abundance from our BioID screen (**C**).
(TIFF)

**S4 Fig. Multiple protein sequence comparison between *Plasmodium spp*. using Clustal Omega alignment.** Highlighted are the bases conserved in all species (black) and respective first amino acids of the proteins (orange).
(TIFF)

**S5 Fig. Generation of *akratin(-)* and *akratin(-)* negative selected parasites via double homologous recombination.** The cartoon shows the cloning strategy and primers used for genotyping. Note that yFCU (Yeast cytosine deaminase-uracil phosphoribosyl transferase fusion protein) was used as a negative selection marker (see methods).
(TIFF)

**S6 Fig. Generation of *P. berghei* and *P. falciparum* complementation parasites via double homologous recombination.** The cartoon shows the cloning strategy and primers used for genotyping.
(TIFF)

**S7 Fig.** (***A***) *Pb*akratin-gfp and *(**B**)* *Pf*akratin-gfp localization in blood and mosquito stages of *P. berghei* parasites. Nuclei (blue) are stained with Hoechst. DIC: differential interference contrast, MG: midgut, SG: salivary gland. Note the weaker *Pf*akratin-GFP signal in gametocytes and the difference between the signals in ookinetes. Scale bars: 5 μm. (**C**) Akratin-GFP localization in activated male and female gametocytes. Scale bars: 5 μm. (**D**) *Pf*akratin-GFP with linker. Localization in blood and mosquito stages. Nuclei (blue) are stained with Hoechst. DIC: differential interference contrast, SG: salivary gland. Scale bar: 5μm. (**E**) Localization of

akratin-GFP in the indicated *P. berghei* life cycle stages. Scale bar: 5 μm. Note that the GFP signal does not correspond to the localization of the intended fusion protein, as GFP is cleaved off.
(TIFF)

**S8 Fig. Gallery of ookinetes expressing *Pb*akratin-GFP, *Pf*akratin-GFP or EYKK/ AAAA-GFP.** Scale bar: 5 μm. Note that the GFP signal does not correspond to the localization of the intended fusion protein.
(TIFF)

**S9 Fig. Generation of *akratin(-)* parasites in the 820line via double homologous recombination.** The cartoon shows the cloning strategy and primers used for genotyping.
(TIFF)

**S10 Fig. Genotyping of *Pf*akratin-mScarlet.** Image shows agarose gel picture of the Integration PCRs. KI = knock-in. Primer combinations are indicated on the right.
(TIFF)

**S11 Fig. Validation and characterization of the *Pf*akratin-TGD line. (A)** Genotyping of *Pf*akratin- TGD. Image shows agarose gel picture of the integration PCRs. Primer combinations are indicated on the right. TGD = targeted gene deletion. **(B)** Gametocyte morphology of three independent experiments. Stages determined by counting them on Giemsa-stained thin blood smears. Normalized to the total number of single-infected RBC. **(C)** Exflagellation of gametocytes in a 5-min window after 12 min activation in four independent experiments. Extended observation time for *Pf*akratin-TGD parasites as indicated. Exflagellation was assessed in technical duplicates on four subsequent days as indicated. Dots represent individual data points. **(D)** Example images for the WGA-based egress assay in (E). Top panels show an exflagellating male gametocyte with flagella and an enlarged nucleus. Scale bar: 5 μm. **(E)** Egress of stage V gametocytes on day 14 normalized to the total number of gametocytes observed. Counting based on shape and presence/loss of the WGA signal of the RBCM. Numbers within the bars indicate the number of gametocytes counted.
(TIFF)

**S12 Fig. Generation of akratin[EYKK/AAAA] parasites via double homologous recombination.** The cartoon shows the cloning strategy and primers used for genotyping.
(TIFF)

**S1 Table. Proteomic analyses of ookinete micronemes.**
(PDF)

**S2 Table. Primer sequences.**
(TIFF)

**S3 Table. Raw data.**
(XLSX)

## Acknowledgments

We thank Miriam Reinig for *Anopheles stephensi* mosquito rearing, Gunnar Mair for help with initial experiments on SOAP-BirA*-tagging and fruitful discussions, Blandine Franke-Fayard and Chris Janse for the 820 line and Markus Ganter for comments on the manuscript. Mass spectrometry and proteomics analysis was performed at the *CellNetworks*-ZMBH core facility at Heidelberg University. We especially thank Bernd Hessling for his support. We also thank

Carolina Barillas-Mury for helpful discussions on ookinete midgut traversal. We acknowledge support from the Infectious Diseases Imaging Platform (IDIP) at the Center for Integrative Infectious Disease Research. FF was a member of the *CellNetworks* Cluster of Excellence (EXC 81) at Heidelberg University. DR, LS, EP and JH were members of the master programs of Molecular Biotechnology (DR) or Infectious Diseases at Heidelberg University.

## Author Contributions

**Conceptualization:** Jessica Kehrer, Tim Gilberger, Friedrich Frischknecht.

**Data curation:** Jessica Kehrer.

**Formal analysis:** Jessica Kehrer.

**Funding acquisition:** Tim Gilberger, Friedrich Frischknecht.

**Investigation:** Jessica Kehrer, Emma Pietsch, Dominik Ricken, Léanne Strauss, Julia M. Heinze.

**Methodology:** Jessica Kehrer.

**Project administration:** Tim Gilberger, Friedrich Frischknecht.

**Supervision:** Jessica Kehrer, Friedrich Frischknecht.

**Visualization:** Jessica Kehrer.

**Writing – original draft:** Friedrich Frischknecht.

**Writing – review & editing:** Jessica Kehrer, Tim Gilberger, Friedrich Frischknecht.

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
