## [Decision Letter · Decision Letter 0]

1 Oct 2024

Dear Dr. Frischknecht,

Thank you very much for submitting your manuscript "APEX-based proximity labeling in Plasmodium identifies a membrane protein with dual functions during mosquito infection" for consideration at PLOS Pathogens. As with all papers reviewed by the journal, your manuscript was reviewed by members of the editorial board and by several independent reviewers. In light of the reviews (below this email), we would like to invite the resubmission of a significantly-revised version that takes into account the reviewers' comments.

We obtained a review from one of the original reviewers from when the paper was first submitted to Review Commons. As you will see from the comments below, the reviewer was satisfied with most of the revisions you provided and agreed that the paper is much improved. However, he/she has additional questions about the subcellular localization of akatrin, specifically about whether it is present in micronemes, which are not thought to be present in gametocytes. This raises the puzzle of where a presumed microneme protein localizes in gametocytes and how its loss causes an exflagellation phenotype. These questions could be addressed with simple localization experiments and/or a more complete discussion of this topic in the text of the manuscript. 

We cannot make any decision about publication until we have seen the revised manuscript and your response to the reviewers' comments. Your revised manuscript is also likely to be sent to reviewers for further evaluation.

Sincerely,

Kirk W. Deitsch

Academic Editor

PLOS Pathogens

Margaret Phillips

Section Editor

PLOS Pathogens

Michael Malim

Editor-in-Chief

PLOS Pathogens

orcid.org/0000-0002-7699-2064

Reviewer's Responses to Questions

**Part I - Summary**

Reviewer #1: In this study the authors use a rapid biotynilation method to identify putative novel microneme proteins in Plasmodium berghei. After filtering of the hits and assessment of false discovery rates, they focus on a top candidate akatrin, and show that deletion of this protein causes a major defect in transmission, with a major effect in male gamete exflagellation.

**Part II – Major Issues: Key Experiments Required for Acceptance**

Reviewer #1: While I am happy with most of the responses of the authors, I still have a concern that would be good to clarify: in my limited understanding, there are no micronemes in gametocytes? And the authors have not provided co-localisation proof that akatrin is in the micronemes. Could you not use the Pf line to show this? This should not take too long and buy confidence that akatrin is used as a marker to validate the APEX data.

If it is not a microneme protein (which would be unlucky, but it could be), the phenotype data still is interesting. But the APEX data would then need to much more critically discussed. While you identify a lot of the top hits as known microneme proteins, there are also many false positives in there. You could probably try to find unifying features of the false positives (peptide count is already being one) that allows you to further increase confidence?

**Part III – Minor Issues: Editorial and Data Presentation Modifications**

Reviewer #1: The discussion currently lacks comments on why a microneme protein (which are normally not present in gametocytes as far as I know) shows a phenotype in exflagellation. I think this would be important to discuss.

PLOS authors have the option to publish the peer review history of their article (what does this mean?). If published, this will include your full peer review and any attached files.

Reviewer #1: No
---

## [Editor Report · Decision Letter 1]

27 Nov 2024

Dear Dr. Frischknecht,

We are pleased to inform you that your manuscript 'APEX-based proximity labeling in Plasmodium identifies a membrane protein with dual functions during mosquito infection' has been provisionally accepted for publication in PLOS Pathogens.

Best regards,

Kirk W. Deitsch

Academic Editor

PLOS Pathogens

Margaret Phillips

Section Editor

PLOS Pathogens

Michael Malim

Editor-in-Chief

PLOS Pathogens

orcid.org/0000-0002-7699-2064
---

## [Editor Report · Acceptance letter]

11 Dec 2024

Dear Dr. Frischknecht,

We are delighted to inform you that your manuscript, "APEX-based proximity labeling in Plasmodium identifies a membrane protein with dual functions during mosquito infection," has been formally accepted for publication in PLOS Pathogens.

Best regards,

Sumita Bhaduri-McIntosh

Editor-in-Chief

PLOS Pathogens

orcid.org/0000-0003-2946-9497

Michael Malim

Editor-in-Chief

PLOS Pathogens

orcid.org/0000-0002-7699-2064